# Patient-initiated second medical consultations—patient characteristics and motivating factors, impact on care and satisfaction: a systematic review

Geva Greenfield [1], Liora Shmueli,[2] Amy Harvey,[1] Harumi Quezada-Yamamoto,[1] Nadav Davidovitch,[3] Joseph S Pliskin,[3] Salman Rawaf [1], Azeem Majeed [1], Benedict Hayhoe [1]

¹Primary Care and Public Health, School of Public Health, Imperial College London Department of Life Sciences, London, UK
²Department of Management, Bar-Ilan University, Ramat-Gan, Israel
³Health Policy and Management, School of Public Health, Ben Gurion University of the Negev, Beer-Sheva, Israel

**Correspondence to**
Dr Geva Greenfield;
g.greenfield@ic.ac.uk

## ABSTRACT

**Objectives** To review the characteristics and motivations of patients seeking second opinions, and the impact of such opinions on patient management, satisfaction and cost effectiveness.

**Data sources** Embase, Medline, PsycINFO and Health Management Information Consortium (HMIC) databases.

**Study design** A systematic literature search was performed for terms related to second opinion and patient characteristics. Study quality was assessed using the National Institutes of Health Quality Assessment Tool for Observational Cohort and Cross-Sectional Studies.

**Data collection/extraction methods** We included articles focused on patient-initiated second opinions, which provided quantitative data on their impact on diagnosis, treatment, prognosis or patient satisfaction, described the characteristics or motivating factors of patients who initiated a second opinion, or the cost-effectiveness of patient-initiated second opinions.

**Principal findings** Thirty-three articles were included in the review. 29 studies considered patient characteristics, 19 patient motivating factors, 10 patient satisfaction and 17 clinical agreement between the first and second opinion. Seeking a second opinion was more common in women, middle-age patients, more educated patients; and in people having a chronic condition, with higher income or socioeconomic status or living in central urban areas. Patients seeking a second opinion sought to gain more information or reassurance about their diagnosis or treatment. While many second opinions confirm the original diagnosis or treatment, discrepancies in opinions had a potential major impact on patient outcomes in up to 58% of cases. No studies reported on the cost effectiveness of patient initiated second opinions.

**Conclusions** This review identified several demographic factors associated with seeking a second opinion, including age, gender, health status, and socioeconomic status. Differences in opinion received, and in the impact of change in opinion, varies significantly between medical specialties. More research is needed to understand the cost effectiveness of second opinions and identify patient groups most likely to benefit from second opinions.

## STRENGTHS AND LIMITATIONS OF THIS STUDY

⇒ This review provides an up-to-date summary of the scientific literature on patient-initiated second medical consultations and adds to a previous review in its breadth.

⇒ The main challenge was in interpretation of findings from different countries with different healthcare systems and different health insurance models.

⇒ Searching for articles in the English-language only means that eligible articles in other languages may have been missed.

## BACKGROUND

A second medical opinion (SO) is a medical decision-making tool for patients, physicians, hospitals and insurers. For patients, it is a way to gain an additional opinion on a diagnosis, treatment or prognosis from another physician.[1] Physicians seeking another colleague's opinion may refer a patient to another consultant to gain further advice. Many health insurers mandate SO programmes to reduce medical costs and eliminate ineffective or suboptimal treatments.[2 3] Hospitals may also require second reviews as part of routine pathology, radiology reviews or for legal purposes. consultant to consultant referrals. Patients in primary care may also request an opinion from a second specialist when unhappy with the opinion from the first specialist.

The clinical impact of insurer-initiated or hospital-initiated second reviews on diagnosis is well documented.[4–8] The value of SOs in pathology and radiology is also well documented, with improvements in the quality of care and reductions in the rate of diagnostic error firmly established.[5–8] The cost effectiveness of routine and mandatory SO programmes has similarly been extensively

studied.[2] [4] However, the cost effectiveness of patient-initiated SOs, and the reasons for initiating SOs, currently remain unclear.

In the context of rising pressure on primary and secondary care services, it is important to set up clear mechanisms for patients seeking second opinions in both public and private systems.

As many patients seek a SO before committing to a treatment plan or a surgery, it is important to understand the advantages versus disadvantages of patient-initiated SOs for themselves, physicians, health services and insurers.[9–11] Seeking a SO may benefit patients medically, provided that the SO is of equal or better quality than the first opinion (FO).[12] Diagnostic errors, thought to occur in 10%–15% of cases in general medicine, may be reduced as a result, and better treatment may be recommended.[13–15] SOs may also benefit patients psychologically by enabling them take control of their care and by offering reassurance.[16] However, it is possible that many SOs do not yield medical benefits for patients and may critically delay the treatment.[12] Likewise, SOs may result in disappointment, confusion or increased uncertainty for patients. SOs may increase physician workload and might be perceived as signalling a patient's distrust, harming the doctor–patient relationship.[16] The cost effectiveness of patient-initiated SOs has also been questioned; SOs may be costly if they involve additional consultations and diagnostic testing, or more expensive treatment recommendations.[4 16 17] In contrast, others have argued that SOs may reduce costs by preventing unnecessary treatment,[4] which is a the rationale for insurer-mandated SOs.

A previous systematic review aimed to determine the clinical outcomes of patient-initiated SOs in general medical and surgical care, their satisfaction, characteristics and motivating factors for seeking SO.[18] The review reported that a surprising paucity of studies have examined the impact of patient-initiated SOs. Patients seeking a SO were mostly women with an average age of 54 years and a diagnosis of breast cancer. Generally, patients were satisfied with SOs, which were more often driven by emotional factors than by concern about their own clinical outcomes. Common motivating factors for seeking a SO were having unresolved symptoms and treatment complications, dissatisfaction with their initial doctor, or seeking additional information. Overall, most patients perceived SOs to be valuable, either because of reassurance or the identification of an alternative.[18] Two other systematic reviews focus on SOs in oncology.[12 19]

As new evidence has been accumulated since the last review, conducted in 2013,[18] we carried out an updated review. We aimed to summarise evidence on (1) the characteristics and motivating factors of patients who initiate SOs; (2) the impact of patient-initiated SOs on diagnosis, treatment, prognosis and patient satisfaction; and (3) their cost effectiveness.

## METHODS

### Eligibility criteria

A systematic review was performed following the Cochrane Handbook for Systematic Reviews of Interventions approach and using the Preferred Reporting Items for Systematic Reviews and Meta-Analyses (PRISMA) statement to report findings.[20 21] A second medical opinion was defined as a situation in which a patient, after getting a medical opinion from one doctor, obtained another opinion from another doctor regarding their diagnosis, treatment, or prognosis. Eligible studies were published in English-language scientific journals with patient-initiated SOs as the focus, which provided quantitative data on their impact on diagnosis, treatment, prognosis or patient satisfaction, described the characteristics or motivating factors of patients who initiated a SO, or analysed the cost effectiveness of patient-initiated SOs. Studies that evaluated only physician-initiated referrals, mandatory or routine second reviews, SOs for legal reasons, online or over-the-phone SOs, or SOs in specialised domains such as dentistry and psychiatry, were excluded. Case studies, conference abstracts, comments, editorials, books and review articles were excluded.

### Information sources

A systematic literature search of Embase, Medline, PsycINFO and HMIC databases was performed. Search terms were keywords related to 'SO' and 'patient'. The search strategy was: ((second adj2 opinion*) OR (second adj2 consult*)) AND patient*. The search strategy was developed with a specialist research librarian at Imperial College London and was deliberately designed to achieve high sensitivity. Additional records were identified through hand searching (of reference lists of relevant papers). No date restriction was applied. The searches were conducted in December 2019.

### Study selection

The records identified through database searching and hand searching were first deduplicated. The titles and abstracts of the remaining records were then independently reviewed by two reviewers (AH and BH) to identify those meeting the inclusion criteria. Ten per cent of the reviewed records were reviewed by another author (GG). Finally, the full text of eligible articles was independently reviewed by two reviewers (AH and BH). Eligibility differences throughout screening were reconciled through discussions.

### Data extraction and quality assessment

A data extraction form was developed and used to capture data elements. Study quality was assessed by AH, BH and GG using the National Institutes of Health (NIH) Quality Assessment Tool for Observational Cohort and Cross-Sectional Studies, with 14 questions being answered for each study.[22] The NIH National Heart, lung and blood institute Quality Assessment Tool for Observational Cohort and Cross-sectional studies and Case Control studies is an established and widely used quality assessment tool. It was deemed appropriate because all included studies employed

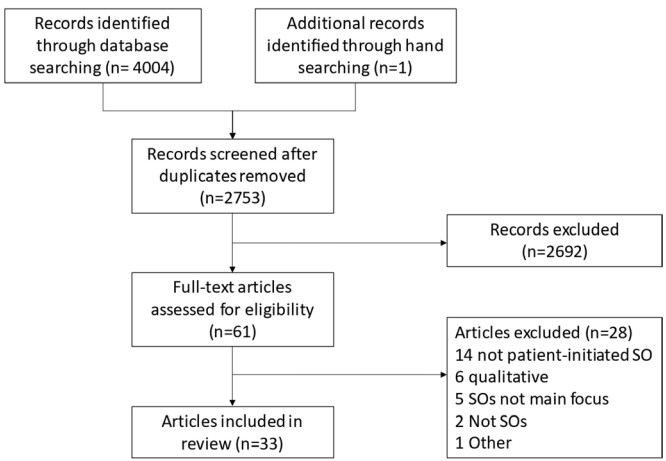

**Figure 1** Preferred Reporting Items for Systematic Reviews and Meta-Analyses flow chart.

an observational study design, to which this quality assessment tool is applicable. The criteria on the NIH Quality Assessment Tool are designed to help researchers focus on the key concepts for evaluating the internal validity of a study.

## Data synthesis and analysis
Evidence tables were constructed detailing the characteristics, medical specialties, results and quality of the studies. The outcome measures were then summarised.

## Patient and public involvement
Patients or the public were not involved in the design, or conduct, or reporting, or dissemination plans of this research.

## RESULTS
Database searching identified 4004 records and hand searching identified one additional record (figure 1). One thousand two hundred fifty-two records were excluded during deduplication, resulting in 2753 unique records. Of these, 2692 were excluded during title and abstract screening, leaving 61 potentially relevant articles. Twenty-eight articles were excluded during a full-text review; 33 articles were included in this review.

## Study characteristics
The 33 included articles described patients with cancer (n=17) and other medical domains (n=16) such as ophthalmology, orthopaedics, neurology and gastroenterology (online supplemental appendix 1). Studies were performed in the USA (n=10), Netherlands (n=7), Israel (n=5), Australia (n=2), Germany (n=3), Japan (n=2), Canada (n=2), Hong Kong and Scotland (both n=1). The 33 studies all used an observational design, either cross-sectional (n=30) or cohort (n=3). The sample size ranged between 36 to 208 366. Studies reported on patient characteristics (n=29), patient motivating factors (n=19), patient satisfaction (n=10) and clinical outcome

agreement (n=17). Detailed study findings appear in online supplemental appendix 2.

## Risk of bias across studies
All studies used an observational design without control patients. All clearly defined their objective, study population, and exposure and outcome measures, and all consistently implemented across all study participants the inclusion and exclusion criteria for participation in the study, and the exposure and outcome measures. However, only 3 studies measured the exposures of interest prior to the outcomes being measured, and only 11 studies measured key potential confounding variables and adjusted them statistically for their impact on the relationship between exposures and outcomes. No studies blinded outcome assessors to the exposure status of participants. No studies evaluated the possibility of the SO having unintended consequences. The sample size of studies was often small, with 23 studies including fewer than 500 participants and 6 including under 100 (online supplemental appendix 3).

## Patient characteristics associated with SO seeking
More females than males had sought a SO: among patients with general medical concerns 52%–61% of patients who sought a SO were female.[11 17 23–29] Three studies conducted in Netherlands, USA, Australia, reported that among patients with cancer 77%–87% of patients who sought a SO were female.[30–32] Conversely, two studies conducted in Japan and Germany reported that more male patients with cancer sought a SO than female.[26 33]

Patients with a higher educational level sought SO more frequently.[1 24 26–28 34–40] Most patients seeking a SO were middle aged. The mean age of patients who sought a SO ranged from 49 to 59 years.[41] The mean age of patients with general medical concerns ranged from 44 to 63 years.[23 25 27 29 42 43]

Seeking a SO was more common in non-religious patients vs religious patients having cancer in Israel,[37] in patients having cancer who were employed in the USA[35 40] and in patients with higher income and socioeconomic status.[1 11 24 28 37] SOs were more common among patients with breast cancer who had private insurance,[35] and among men with localised prostate cancer with private insurance in the USA.[40] Two studies reported on geographical residency, more common for those living in central areas in Israel[11] and for those closer to a SO centre in the Netherlands.[44] Patients seeking a SO with breast cancer were more actively involved in decision-making processes in Germany.[34] Patients seeking a SO from orthopaedics had a poorer relationship with their first doctor in the Netherland[44] and those seeking SO in Japan were more anxious and believed they were in poor health.[27] Seeking a second opinion was negatively related to internal locus of control, perceived health status, and wish to know all details of treatment.[9]

## Patient motivating factors
The most common reason reported for seeking an SO was to confirm or refute the suggested diagnosis or treatment

or[25 30 32 33 45]; where patients disagreed with their doctor on diagnosis, 44.3% sought a SO.[41] Eighty-five per cent of patients seeking an SO reported on poorly defined problems by their first physician, and 79% for a change in treatment.[26] For example, 59% of patients seeking an SO at a neurological clinic hoped for a different diagnosis or treatment than the first opinion.[29] Among orthopaedic patients, 38%–40% questioned the first diagnosis or believed it was incorrect[44 46] and 18% sought reassurance about a recommended surgery.[47] Forty-one per cent of ophthalmology patients sought an SO because their first doctor indicated that no treatment was possible, or that their prognosis was poor.[23] Patients often sought SOs where they disagreed with their doctor on proposed treatments (29% of drug-related disagreements, and 53% of other treatment disagreements).[41]

Patients often sought an SO to get more information related to diagnosis, treatment options and reassurance.[48] Some were seeking a subspecialist's opinion,[46] with the natural wish 'to be seen by the best doctor'.[39] Dissatisfaction with communication with the first doctor ranged from 19%[46] to 51%,[44] where some believed that the first physician did not spent enough time with them.[45] Some patients were encouraged by family members or friends to seek a SO,[48] or were recommended a certain doctor by family or friends.[44]

### Patient satisfaction

Patients were commonly very satisfied with the SO they received. The SO provided them with reassurance of their treatment or diagnosis, gaining comprehensible information about the treatment,[33 48] with a compassionate approach addressing their needs[48] and obtaining answers to their concerns.[32] Eighty-four per cent of SO seekers among the general adult population in Israel were satisfied with the SO and 91% preferred the SO over the FO.[46] Ninety-five per cent of patients enrolled in a national SO programme in the USA were satisfied with the experience and 87% were more confident in their diagnosis or treatment.[47] In a survey conducted in Japan, most patients who obtained an SO reported they better understood their treatment options (93%), their illness (88%) and the risks of their treatment (82%).[26] SO consultations in neurology received higher scores than the FO consultations across many aspects of satisfaction: patient involvement in the conversation and in decision-making, information and emotional support given.[29] However, during a 2-year follow-up study, overall satisfaction decreased to the same level as before the SO consultation.[49] Out of 37, 21 parents of children with cancer in a paediatric haematology oncology department were satisfied with the second opinion they received.[37]

Most patients in all studies were satisfied with their SO consultation. Patients reported feeling more knowledgeable and reassured about their diagnosis and treatment,[32] and reported their trust in the attending physician was strengthened by getting a second opinion.[33] Some patients believed that the second doctor communicated better, answering

concerns and providing more information (51%), listening more (39%) and being friendlier (41%).[32]

### Clinical agreement between the first and SO

Substantial discrepancies between the first and SOs in diagnosis and suggested treatment were reported across the studies. Diagnosis was confirmed in 50%[26] to 57%[47] of cases, clarified in 17% and changed in 13%[26] to 15%.[47] Among women seeking an SO at a uterine fibroid treatment centre, 13.2% of previous diagnoses of uterine fibroids were unconfirmed by the SO.[43] In people who sought an SO for general medical concerns while enrolled in a national SO programme, diagnosis was confirmed in 56.8% of cases, clarified in 17% and changed in 14.8%.[47] In patients seeking an SO at an eye hospital, there was 67.9% agreement with surgery recommendations between the FO and SO consultations.[23] Changes in both diagnosis and treatment were experienced by 11%[47] to 56%[29] of patients who sought a SO.

Among patients with lung cancer, differences were found between the FO and the SO in 9% of diagnoses (17 patients) and in 13% of cancer stage classification (24 patients) and in 37% of therapeutic advice (70 patients). In total, there were 91 discrepancies between the FO and SO, of which 53 (58%) had a potential major impact on survival, morbidity and quality of life.[50]

In surgical oncological cases where the second and first opinions could be directly compared, the advice was identical in 68%, there was a major discrepancy in 16% and a minor discrepancy in another 16%.[31]

SO treatment recommended for surgical breast cancer deviated from the FO consultation in 20.3% of 54 cases.[51] Thirty-five per cent of 37 parents of children with haematological cancer were advised to change the treatment advised in the FO.[37] However, 56% of patients with breast cancer did not receive a recommendation for surgery either in their FO or SO consultation.[38]

SOs received had a substantial impact on patient decision making. For 42% of patients with cancer, their SO consultation resulted in a change of treatment.[32] Sixty-eight of patients with general medical concerns mentioned they would change or partially change the treatment when the SO and FO differed.[26]

### Cost effectiveness

No studies were found to report on the cost effectiveness of patient-initiated SOs.

## DISCUSSION
### Summary of findings

Women tended to seek SOs more than men. Most patients seeking a SO were middle aged, with a higher educational level. They tended to be employed, have a higher income and socioeconomic status, and have private medical insurance. Patients seeking an SO sought to gain more information about their condition, gain reassurance about their diagnosis or treatment, were dissatisfied with their previous

doctor or were encouraged by family members or friends to seek a SO. Seeking SOs in many cases stemmed from dissatisfaction with the information and the communication with the first doctor, where patients felt they were not given the information or reassurance they sought. Most patients were satisfied with their SO consultation, felt more knowledgeable and reassured about their diagnosis and treatment, and reported having more confidence and trust in their second doctor. Patients believed that their SO doctor communicated better, listened more and was friendlier. A considerable proportion of SO consultations yielded a change in diagnosis or treatment, and these discrepancies had potentially major impact on patient outcomes in up to 58.2% of lung cancer cases. Despite the cost effectiveness of routine and mandatory SO programmes having been extensively studied,[52–54] we found no studies reporting on the cost effectiveness of patient-initiated SOs.

## Strengths and Limitations

The review offers a broad overview on the topic of SOs and adds to the previous review in terms of breadth and up-to-dateness.[18] We designed a high-sensitivity search strategy, which did not rely on the 'referral and consultation' term used in the previous review. This because a second opinion does not necessarily require a referral, and in many healthcare systems there is no gatekeeping for second opinions and patients can contact a physician privately and independently for a second opinion.

Some limitations should be acknowledged. The main challenge in interpreting these findings is in the cohort of studies from different countries and different healthcare systems, where different insurance models are in place. For example, in some countries and under specific insurance schemes, access to SOs is covered by national and private insurers, whereas in other systems, SOs would be out-of-pocket. Comparison between countries is challenging, as there are substantial differences, not just in the country level, as even in the same country there are different healthcare models and insurance models in each country, not to mention cultural differences in attitudes toward second opinions, which play a significant role. Differences in cultures and attitudes towards parallel consultations with different doctors may also affect the findings presented in studies in this review. Likewise, searching only for articles in the English-language means that we may have missed eligible articles in other languages.

## Comparison with previous research

The review offers an updated and broad perspective on patient-initiated SOs. A direct comparison is challenging because we used a different search strategy. This review identified an additional 18 studies, 9 of which were published before the previous review.[18] Three studies[55–57] were included in the previous review[18] but not in this review, because they did not refer to purely patient-initiated SOs;[55 57] hence, the patient behaviour could not be separated from physician-initiated SOs. Another study referred medical nomadism,[56] which is an allied but a different to

a seeking second opinion, since it also includes seeking multiple opinions from different experts, not necessarily from the same area of expertise.

Both reviews included only observational studies with an absence of data on control patients. Both reviews found no studies which evaluated the possibility of the SO having unintended consequences. Regarding the characteristics of patients who had sought a SO, the previous review reported only that a large proportion of patients seeking a SO were women with an average age of 54 with a diagnosis of breast cancer. The education level of SO seekers ranged from those with less than a high school education to those with a university degree. This review referred to a broad range of factors pertaining to religious belief; employment, income and insurance; geographical residency; preference for involvement in decision-making; relationship with their first doctor; anxiety and beliefs they were in poor health.

We found similar motivating factors of patients compared with the previous review, with the vast majority of motivating factors for both patients with cancer and patients with general medical concerns related to gaining more information about their condition, reassurance about their diagnosis or treatment, or dissatisfaction with their previous doctor. Both reviews found most patients in the studies to be satisfied with their SO consultation; however, a cohort study in this review reported that patient satisfaction dropped in the 2 years following the SO consultation to slightly below the satisfaction with the FO consultation. Both reviews found that SOs most typically confirm the original diagnosis or treatment, but that a considerable proportion of SOs yield a change. We also report that some medical specialties experience significantly more or fewer changes in diagnosis or treatment than average, and that changes in diagnosis and treatment have a more significant impact in cancer patients than in patients with general medical concerns. Two other systematic reviews focus specifically on SO in oncology.[12 19] We did not limit to specific medical specialties and so report evidence on SO in all medical domains.

## Implications for practice

While SOs usually confirm the original diagnosis or treatment, a considerable proportion of SO consultations yield a change in treatment. Some medical specialties experienced significantly more changes in diagnosis or treatment, and changes in diagnosis and treatment had a more significant impact in patients with cancer than in patients with general medical concerns. In specialities where there are often major discrepancies, there is a case to initiate a SO systematically or at least to make patients aware of the option of seeking a SO. Likewise, in cases where patients delay or avoid making a decision about a treatment course, SOs can help reassure and expedite the treatment. SO may benefit patients emotionally, even if they do not result in medical changes.

The fact that patients seeking an SO tended to be more educated patients, with higher income or socioeconomic status, having private insurance and living in central urban areas, raise concerns about inequalities and access to SOs

among deprived groups and those living in rural areas, where access to specialists is limited.

While in many cases the SO confirms the FO, from the patient perspective, a change in their diagnosis or a treatment course may have a crucial impact on their lives, particularly in surgical oncology. From the healthcare system or the insurer perspective, changes in diagnosis or treatment, even if they occur in only a portion of patients, may have substantial impact on patient outcomes, rehabilitation, costs and healthcare staff resources. For example, in the NHS in England, there is a legal requirement that every histopathology assessment should be by two pathologists, which is also built in the health system costing.

SOs stemming from unsatisfactory communication with the first doctor could be potentially avoided by improved doctor-patient communication, offering a detailed explanation and a listening approach. Rather than the SOs being sought confidentially, to not offend the first doctor, doctors should encourage a SO if they sense the patient is in doubt and assist in referring the patient to a suitable consultant and help to come to a mutual decision based on a discussion between the patient and both doctors. By negotiating a treatment that is acceptable to all parties, patients may be spared the confusion associated with discrepant opinions. By preparing patients for the various potential positive and negative outcomes of a SO, doctors can help them make an informed decision about pursuing the SO.

More people taking SO in national healthcare systems will put additional strain on the secondary care, but if unnecessary surgery is cancelled following a SO this will release resources, not to mention a long rehabilitation process which often follow surgery.

### Future research

Although our review suggests that patients generally believe SOs to be valuable, studies infrequently presented follow-up data on patient outcomes. It would also be useful to further explore the extent to which patients are referred back to their initial doctor, and to what extent SOs actually changed the course of treatment (rather than the mere fact that an additional opinion had been obtained). There is a distinct lack of studies on the cost-effectiveness of patient-initiated SOs, despite extensive literature on the cost effectiveness of routine and mandatory SO programmes. Long-term outcomes and potential unintended consequences of SOs must also be examined. Likewise, there is a lack of a uniform definition or objective measures of 'SO', which makes the comparison of findings across studies and health systems challenging. Development of uniform measures will be useful to uniformly compared findings across different countries and healthcare systems. The health systems and related insurance models' aspects, while highly relevant, warrant a broader discussion which was beyond the remit of this review.

## CONCLUSIONS

We identified demographic characteristics associated with seeking a second opinion, related to age, gender, education, socioeconomic status, place of residence and health condition. Patients seeking a second opinion sought to gain more information or reassurance about their diagnosis or treatment. While many second opinions confirm the original diagnosis or treatment, discrepancies in opinions had a potential major impact on patient outcomes. Research is needed to examine cost effectiveness of second opinions and to identify patient groups that are likely to benefit from a second opinion. In the context of rising pressure on primary and secondary care services, it is important to set up clear mechanisms for patients seeking second opinions in both public and private systems.

**Contributors** GG, LS and BH were involved with conception and design, conducted the data analysis, and drafted the manuscript. AH and HQ-Y were involved in designing and conducting the literature searches, screening, data extraction and synthesis, and revised various versions of the manuscript. ND, JSP, SR and AM were involved in conception and design, interpretation of the findings, provided clinical perspectives, and revised various versions of the manuscript.

**Funding** This report was supported by the National Institute for Health Research Applied Research Collaboration Northwest London, Award Number NIHR200180.

**Disclaimer** This article presents independent research commissioned by the National Institute for Health Research (NIHR) under the Applied Health Research (ARC) programme for North West London. The views expressed in this publication are those of the author(s) and not necessarily those of the NHS, the NIHR or the Department of Health.

**Competing interests** None declared.

**Patient consent for publication** Not required.

**Provenance and peer review** Not commissioned; externally peer reviewed.

**Data availability statement** Data are available upon reasonable request. All data are publicly available.

**ORCID iDs**
Geva Greenfield http://orcid.org/0000-0001-9779-2486
Salman Rawaf http://orcid.org/0000-0001-7191-2355
Azeem Majeed http://orcid.org/0000-0002-2357-9858
Benedict Hayhoe http://orcid.org/0000-0002-2645-6191

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
