## [Reviewer comments · BMJ Open]

ARTICLE DETAILS

TITLE (PROVISIONAL)	Patient-initiated second medical consultations: patient characteristics and motivating factors, impact on care and satisfaction: A systematic review
AUTHORS	Greenfield, Geva; Shmueli, Liora; Harvey, Amy; Quezada-Yamamoto, Harumi; Davidovitch, Nadav; Pliskin, Joseph; Rawaf, Salman; Majeed, Azeem; Hayhoe, Benedict

VERSION 1 – REVIEW

REVIEWER	Gulbrandsen, Pal Universitetet i Oslo Det medisinske fakultet
REVIEW RETURNED	30-Sep-2020

GENERAL COMMENTS	The paper is relevant, and adds some substantial knowledge compared to the former review on the same subject. I find the results plausible. Comments/questions: 1) Abstract, principal findings. I think a percentage with a comma is too much precision.2) Abstract, conclusions. Personally, I find the conclusion more an opinion than a conclusion. In a world of massive amounts of research papers in several difficult fields, I question the old term "research is needed...". One could raise the question why we need to know more about a type of behavior that hardly can be avoided in countries where people are free to choose what to do for themselves. Basically, this is a matter of trust in the system, high usage mirrors less trust or lack of ability to voice, I suppose. Such discussions are outside the scope of the article, so I would suggest the conclusion just briefly repeats main findings.3) Page 4 has the title Strengths and limitations of this study (which is also a para in the discussion). To me, these three bullet points look more like highlights of the study. If that is what they are meant to be, I consider bullet point two to be too detailed. The sentence in bullet point three is not clear. I agree, however, that the point about different insurance models is highly relevant.4) Background. I think the paper would be strengthened by briefly mentioning possible between country variation in how systems are set up, particularly as this seems to be one of the motivations for the study and implications. One way to approach this is to bring into the text in which countries the referred studies originate.5) Results, para "Risk of bias across studies". The same fact is mentioned twice (Only 3 studies...)6) Results, para "Patient characteristics...". First para here, I would be very interested in where studies with conflicting findings were made. Sociological/cultural differences could account for gender differences.
---

	7) Results, same page, 3rd para: Seeking a SO was common (I miss the word more before common) 8) Patient motivating factors, first para, top of page 9: "Undiagnosed complaints...", this sentence remains unclear. 9) Clinical agreement, last para: I would think the preposition ahead of patient decision making should be on, not of. 10) Discussion. In general, the discussion is OK and limited to the findings of the systematic review, and I find this appropriate. This type of study does not really open up for more principal discussion about the use of patient initiated second opinion, and how systems and societies can handle the costs, risks, and benefits of such behavior. I admit my opinion having read this review is that we need (even) more studies to have better data on these issues. I think these questions are genuinely political, ethical, moral, including quality in health systems. I would support an initiative to a broader discussion to follow this paper, a discussion that should for example include a detailed overview of how different systems deal with the challenge.
--	---

REVIEWER	Li, Shuangyu King's College London, Division of Medicine
REVIEW RETURNED	04-Oct-2020

GENERAL COMMENTS	This systematic literature review synthesises published evidence about patients seeking second opinions (SOs) in their healthcare. It focuses on the characteristics and motivating factors, impact on diagnosis treatment prognosis, patient satisfaction and cost-effectiveness. The review yields an improved understanding of these questions in comparison with other reviews including one by the authors themselves in 2013. The results and discussion may have significant impact on clinical practice, which the authors discuss comprehensively in the article. This review process is well explained, and details well documented in the appendices. This review is an update of their 2013 review. The authors have taken a more focused approach, which fits the purposes as set in the refined research questions. The discussion clearly discusses how the two reviews differ from each other, which highlights the new knowledge the current review adds. I have a concern about the lack of clarify on cross country comparison. While the authors acknowledge in the limitations section that the effect of the healthcare system, insurance and culture on patients' SO seeking behaviour is not explored in this review, I feel some explanations are still needed in order to contextualise the generalisability of the review results. 'Patient satisfaction' is the only section that mentions from which country the results are. I wonder if the same style can apply to other sections. It would be helpful to add some information about how different countries compare with one another in each of the results sections, to help avoid potential misassumption that they are more or less the same (or are they?). There have been a few repetitions about the key findings in Discussion and Conclusion, which is helpful for the readers to remember. However, some of them may be reduced to save space for adding the contextual information. There are some small formatting errors. For example, in line 34, a space is missing between 'female' and '['. There are a few others at the start of the article. Line 16, 'consultation to consultant referrals' is not a complete sentence.
---

REVIEWER	Gerwing, Jennifer Akershus University Hospital, Health Services Research
REVIEW RETURNED	06-Oct-2020

GENERAL COMMENTS	BMJopen-2020-044033 The authors aim to review the literature focused on patients initiating second opinions, specifically on characteristics of patients who do so, factors that motivate patients to seek a second opinion, and the impact of seeking a second opinion (on diagnosis, prognosis, treatment, patient satisfaction, and cost-effectiveness). The authors use a general keyword search strategy on Embase, Medline, PsycINFO, and HMIC databases and select 31 studies for review. The findings match the heterogeneity of the clinical settings, countries, and health care systems; no findings are provided on cost effectiveness. The article is generally well written, with a few editing issues (e.g., repeated sentences page 8, lines 13-17 and lines 21-25). The authors could consider removing “cost effectiveness” from the title, given that no papers were found that could provide results on that topic. The motivation driving the study is primarily updating Payne et al.’s 2014 systematic review on patient-initiated second opinions. I am concerned that the current manuscript’s summary of the 2014 review is not entirely accurate: For example, p. 5, starting at line 53, the former study is characterized as being focused on “general medical care”; however, Payne et al. covered studies from oncology, elective surgery, and general medical care. Also, page 6, paragraph 2 (lines 13-24) characterizes (by implication) the previous review as less refined; however, the former authors had used a variety of Medical Subject Headings (MeSH) indexes and keyword searches. The authors of the present manuscript used a simpler search strategy, reported in Appendix 1 as a one-line Boolean operator. The current manuscript could be improved by involving a librarian with specialized skills related to searching, source selection, planning, formulating questions. I would be curious to read more about the discrepancies in papers included between the two systematic reviews, which overlap in only 8 papers, with the previous (2014) review including 5 studies not included in the present review. Discounting the papers from 2013 on, the present review included 7 papers not included in the previous review. This difference is interesting, and the authors could speculate on the disagreement in more detail. As the authors mention, the 31 studies are observational and thus risk low quality and high bias. Did the authors do a bias check (e.g., using the Cochrane Risk of Bias Tool)? The authors report a quality check using the NIH Quality Assessment Tool for Observational Cohort and Cross-sectional Studies, and they describe risks (p. 8 first paragraph), but it could be useful to report the raters’ actual quality rating (Good, Fair, or Poor) for each study, or at least a summary of these ratings and how any disagreements between raters were resolved. The results are nicely organized and succinct. They would benefit from a Table summarizing the specific findings from each study in more detail. One concern is the heterogeneity of the studies (which the authors also point out), with variability in the original research questions, clinical settings, country in which the study was conducted, and health care systems. It would be useful, therefore, to have each
---

	result contextualized by this information, particularly when results are from a single study, which could make them appear more generalizable than they are. For example, the result that “seeking a SO was common in non-religious patients vs. religious patients” would be enhanced with information about the setting (pediatric oncology) and country (Israel). I also question the accuracy of some of the results. For example, I struggled to reconcile the findings in the manuscript with the Annadale and Hunt (1998) study. On p. 8, lines 39-41, there is a reported age result in the cancer setting. The original paper, however, drew on a sample of participants from a 35-year-old age cohort, there were no age-related findings, and there were a variety of health complaints. Further, in Table 1 this study is reported as having 136 participants, when in fact there were 307 cases that included disagreements, of which 136 reported seeking a second opinion. How patients are engaged in decision making is a focus of much current scholarship, and the authors have undertaken an ambitious review related to this important topic: why do patients seek second opinions, under what conditions do they do so, and what outcomes result? The manuscript would benefit from increased rigor in method and accuracy of results, which could then contribute a valuable update to the 2014 systematic review.
--	--

REVIEWER	Chen, Yan University of Auckland, Centre for Medical and Health Sciences Education
REVIEW RETURNED	04-Feb-2021

GENERAL COMMENTS	This paper is well written and describes the study methods and findings in great detail. I was invited to review the paper with a particular emphasis on its statistical methods and analyses. The manuscript follows the PRISMA guideline and provides sufficient information to adhere to the guideline. I have very few minor comments regarding the study’s analyses, given that it has demonstrated sound methodology. In the eligibility criteria, the authors excluded subspecialized domains such as dentistry and psychiatry; whereas some other specialities such as neurology, ophthalmology, and orthopaedics were included. It would have been beneficial to explain why psychiatry, as a medical sub-specialty, was being excluded. There are some inconsistencies in-text. For instance, the authors referred to Appendix 2 in-text, while I assume that they were actually referring to Table 2 instead.
--

VERSION 1 – AUTHOR RESPONSE

Reviewer 1

Dr. Pal Gulbrandsen, Universitetet i Oslo Det medisinske fakultet

Comments to the Author:

The paper is relevant, and adds some substantial knowledge compared to the former review on the same subject. I find the results plausible.

Comments/questions:

Abstract, principal findings. I think a percentage with a comma is too much precision.

We would like to thank you for taking the time to review the paper and provide helpful comments, which improve the quality for manuscript.

The percentage has been changed to an integer presentation.

2) Abstract, conclusions. Personally, I find the conclusion more an opinion than a conclusion. In a world of massive amounts of research papers in several difficult fields, I question the old term "research is needed...". One could raise the question why we need to know more about a type of behavior that hardly can be avoided in countries where people are free to choose what to do for themselves. Basically, this is a matter of trust in the system, high usage mirrors less trust or lack of ability to voice, I suppose. Such discussions are outside the scope of the article, so I would suggest the conclusion just briefly repeats main findings.

Thank you for this suggestion. We have revised the conclusions section in the abstract to summarise the main findings as follows:

"The results of this review contribute to comprehensive information on the characteristics and motivations of patients seeking second opinions. Our findings show that seeking a second opinion was more common in women, middle age patients, more educated patients; and in people having a chronic condition, with higher income or socioeconomic status or living in central urban areas. Patients seeking a second opinion sought to gain more information or reassurance about their diagnosis or treatment".

3) Page 4 has the title Strengths and limitations of this study (which is also a para in the discussion). To me, these three bullet points look more like highlights of the study. If that is what they are meant to be, I consider bullet point two to be too detailed.

The sentence in bullet point three is not clear. I agree, however, that the point about different insurance models is highly relevant.

We have revised the "strengths and limitations" section as follows, to focus more on methodological strengths and limitations:

- This review provides an up-to-date summary of the scientific literature on patient-initiated second medical consultations and adds to a previous review in its breadth

- The main challenge was in interpretation of findings from different countries with different healthcare systems and different health insurance models
- Searching for articles in the English-language only means that eligible articles in other languages may have been missed.

4) Background. I think the paper would be strengthened by briefly mentioning possible between country variation in how systems are set up, particularly as this seems to be one of the motivations for the study and implications. One way to approach this is to bring into the text in which countries the referred studies originate.

The countries from which the studies originated are described in Table 1 and have been now described in the text where applicable (Particularly in the “Patient Characteristics associated with SO Seeking” section in the Results). Bearing in mind the original aim of this study and the word limit, we realised it would be challenging to provide a summary of country variation in terms of the way systems are setup in sufficient detail, as there are so many characteristics for each healthcare system which might affect second opinion seeking. The main purpose was to review the characteristics and motivations of patients seeking second opinions, and the impact of such opinions on patient management, satisfaction, and cost-effectiveness. The health systems and related insurance models’ aspects are beyond the remit of this review.

5) Results, para "Risk of bias across studies". The same fact is mentioned twice (Only 3 studies...)

We have deleted the double sentence under the "Risk of bias across studies" paragraph.

6) Results, para "Patient characteristics...:". First para here, I would be very interested in where studies with conflicting findings were made. Sociological/cultural differences could account for gender differences.

We have now added the location of the studies in which conflicting findings were identified, e.g. for gender under the results section, paragraph “Patient Characteristics associated with SO Seeking as follows: “Three studies conducted in Netherlands, US, Australia reported that among cancer patients 77-87% of patients who sought a SO were female [32–34]. Conversely, two studies conducted in Japan and Germany reported that more male cancer patients sought a SO than female [27,35]”.

7) Results, same page, 3rd para: Seeking a SO was common (I miss the word more before common)

We have revised the following sentence:

“Seeking a SO was more common in non-religious patients vs. religious patients.”

8) Patient motivating factors, first para, top of page 9: "Undiagnosed complaints...", this sentence remains unclear.

We have revised the following sentence:

“85% of patients seeking a SO reported on poorly defined complaints by their first physician”.

9) Clinical agreement, last para: I would think the preposition ahead of patient decision making should be on, not of.

We have revised the following sentence:

“SOs received had a substantial impact on patient decision making.”

10) Discussion. In general, the discussion is OK and limited to the findings of the systematic review, and I find this appropriate. This type of study does not really open up for more principal discussion about the use of patient initiated second opinion, and how systems and societies can handle the costs, risks, and benefits of such behavior. I admit my opinion having read this review is that we need (even) more studies to have better data on these issues. I think these questions are genuinely political, ethical, moral, including quality in health systems. I would support an initiative to a broader discussion to follow this paper, a discussion that should for example include a detailed overview of how different systems deal with the challenge.

We fully agree that we need more studies on these issues. Indeed, we have published a study pertaining to this issue (Shmueli et al. Second opinion utilization by healthcare insurance type in a mixed private-public healthcare system: a population-based study. *BMJ Open*. 2019 Jul 27;9(7):e025673. doi: 10.1136/bmjopen-2018-025673.).

Comparison between countries is challenging. There are substantial differences between health systems in different countries, and even within countries there are different healthcare models and insurance models, not to mention cultural differences in attitudes toward second opinions, which play a significant role. Some countries operate a fully public model, some a fully private model and some a mixed public-private model. Hence any comparison should consider not only health systems on a country level, but between specific models within the same country. Likewise, health systems characteristics are often extremely complicated and comparing patient behaviour by country requires adjustment for multiple confounding variables on the country level. Such analysis was outside of the scope of this paper, which was to review the characteristics and motivations of patients seeking

second opinions, and the impact of such opinions on patient management, satisfaction, and cost-effectiveness. The health systems and related insurance models' aspects, while highly relevant, warrant a broader discussion which was beyond the remit of this review.

We have added though a shorter version of the explanation given here to the discussion.

Reviewer: 2

Dr. Shuangyu Li, King's College London

Comments to the Author:

This systematic literature review synthesises published evidence about patients seeking second opinions (SOs) in their healthcare. It focuses on the characteristics and motivating factors, impact on diagnosis treatment prognosis, patient satisfaction and cost-effectiveness. The review yields an improved understanding of these questions in comparison with other reviews including one by the authors themselves in 2013. The results and discussion may have significant impact on clinical practice, which the authors discuss comprehensively in the article.

This review process is well explained, and details well documented in the appendices. This review is an update of their 2013 review. The authors have taken a more focused approach, which fits the purposes as set in the refined research questions. The discussion clearly discusses how the two reviews differ from each other, which highlights the new knowledge the current review adds.

I have a concern about the lack of clarify on cross country comparison. While the authors acknowledge in the limitations section that the effect of the healthcare system, insurance and culture on patients' SO seeking behaviour is not explored in this review, I feel some explanations are still needed in order to contextualise the generalisability of the review results. 'Patient satisfaction' is the only section that mentions from which country the results are. I wonder if the same style can apply to other sections. It would be helpful to add some information about how different countries compare with one another in each of the results sections, to help avoid potential misassumption that they are more or less the same (or are they?).

We would like to thank you for taking the time to review the paper and provide helpful comments, which improve the quality for manuscript.

We fully agree that we need more studies on these issues. Indeed, we have published a study pertaining to this issue (Shmueli et al. Second opinion utilization by healthcare insurance type in a mixed private-public healthcare system: a population-based study. *BMJ Open*. 2019 Jul 27;9(7):e025673. doi: 10.1136/bmjopen-2018-025673.).

Comparison between countries is challenging. There are substantial differences between health systems in different countries, and even within countries there are different healthcare models and insurance models, not to mention cultural differences in attitudes toward second opinions, which play a significant role. Some countries operate a fully public model, some a fully private model and some a mixed public-private model. Hence any comparison should consider not only health systems on a country level, but between specific models within the same country. Likewise, health systems characteristics are often extremely complicated and comparing patient behaviour by country requires adjustment for multiple confounding variables on the country level. Such analysis was outside of the scope of this paper, which was to review the characteristics and motivations of patients seeking second opinions, and the impact of such opinions on patient management, satisfaction, and cost-effectiveness. The health systems and related insurance models' aspects, while highly relevant, warrant a broader discussion which was beyond the remit of this review.

We have added though a shorter version of the explanation given here to the discussion.

There have been a few repetitions about the key findings in Discussion and Conclusion, which is helpful for the readers to remember. However, some of them may be reduced to save space for adding the contextual information.

We have now tightened the conclusion section with a more succinct summary of the findings.

There are some small formatting errors. For example, in line 34, a space is missing between 'female' and '['. There are a few others at the start of the article.

We have reviewed and corrected the spacing throughout the article.

Line 16, 'consultation to consultant referrals' is not a complete sentence.

We have revised the following sentence:

"Physicians seeking another colleague's opinion may refer a patient to another consultant to gain further advice."

Reviewer: 3

Ms. Jennifer Gerwing, Akershus University Hospital

Comments to the Author:

BMJopen-2020-044033

The authors aim to review the literature focused on patients initiating second opinions, specifically on characteristics of patients who do so, factors that motivate patients to seek a second opinion, and the impact of seeking a second opinion (on diagnosis, prognosis, treatment, patient satisfaction, and cost-effectiveness). The authors use a general keyword search strategy on Embase, Medline, PsycINFO, and HMIC databases and select 31 studies for review. The findings match the heterogeneity of the clinical settings, countries, and health care systems; no findings are provided on cost effectiveness.

The article is generally well written, with a few editing issues (e.g., repeated sentences page 8, lines 13-17 and lines 21-25).

The authors could consider removing “cost effectiveness” from the title, given that no papers were found that could provide results on that topic.

We would like to thank you for taking the time to review the paper and provide helpful comments, which improve the quality for manuscript.

We have revised the title as follows: “Patient-initiated second medical consultations: patient characteristics and motivating factors, impact on care and satisfaction: A systematic review.”

The motivation driving the study is primarily updating Payne et al.’s 2014 systematic review on patient-initiated second opinions. I am concerned that the current manuscript’s summary of the 2014 review is not entirely accurate: For example, p. 5, starting at line 53, the former study is characterized as being focused on “general medical care”; however, Payne et al. covered studies from oncology, elective surgery, and general medical care. Also, page 6, paragraph 2 (lines 13-24) characterizes (by implication) the previous review as less refined; however, the former authors had used a variety of Medical Subject Headings (MeSH) indexes and keyword searches.

We have revised the sentence to “in general medical and surgical care”.

We did not seek critically to review the previous study in this manuscript. However, in the previous review, only eight of the thirteen studies contained data on patient initiated SOs. Two studies referred to doctor-shopping behaviour and to medical nomadism (where patients consult with multiple doctors for the same symptomatology during a certain period), which are different help-seeking behaviours than seeking a SO in terms of patient profile and motivation for seeking further advice. In three studies, data on patient-initiated SOs could not be separated from physician-initiated SOs. We aimed to overcome these limitations in this review (please see our response to another comment below).

The section has been revised accordingly.

The authors of the present manuscript used a simpler search strategy, reported in Appendix 1 as a one-line Boolean operator. The current manuscript could be improved by involving a librarian with specialized skills related to searching, source selection, planning, formulating questions.

Our group is highly experienced with sophisticated database searches in systematic reviews and has published systematic reviews extensively. As an active group in second opinions research over the last 15 years we are highly knowledgeable on the subject matter. The reason for the simplified search strategy on this occasion was not oversight or insufficient knowledge of how to construct complicated search strategies, but a deliberate decision made together with a highly experienced specialist research librarian to keep a high-sensitivity search strategy. This was combined with careful screening work later on during the title and abstract and full text screening phases. The search strategy employed in the previous review is indeed lengthier however, it relied heavily on the “Referral and Consultation” term which might have little relevance to second opinions. For this reason, we did not include it in the search strategy.

I would be curious to read more about the discrepancies in papers included between the two systematic reviews, which overlap in only 8 papers, with the previous (2014) review including 5 studies not included in the present review. Discounting the papers from 2013 on, the present review included 7 papers not included in the previous review. This difference is interesting, and the authors could speculate on the disagreement in more detail.

Further to our response to the previous comment, our search strategy was deliberately designed to achieve high sensitivity, which likely resulted in identification of studies which were not found by the previous review. The over reliance on the “referral and consultation” term in the previous review might have limited the identification of relevant studies, as a second opinion does not necessarily require a referral, and in many healthcare systems there is no gatekeeping for second opinions and patients can contact a physician privately and independently for a second opinion.

Considering the five studies which were included in the previous review but not in this review, Grafe 1978 was not on patient-initiated SO, Bekkelund et al. looked in our interpretation at first opinions, while Boudali et al. referred to medical nomadism, which is related but different to a seeking second opinion, since it also includes more than two opinions from different experts, not necessarily from the same area of expertise, and so does not conform to a definition of second opinion.

However, Sutherland 1989 and 1994, do appear to have been screened out during the title and abstract phase inappropriately. We have now re-evaluated them and included them in the review.

We have added this description to the “Comparison with Previous Research” section in the discussion.

As the authors mention, the 31 studies are observational and thus risk low quality and high bias. Did the authors do a bias check (e.g., using the Cochrane Risk of Bias Tool)? The authors report a quality check using the NIH Quality Assessment Tool for Observational Cohort and Cross-sectional Studies, and they describe risks (p. 8 first paragraph), but it could be useful to report the raters' actual quality rating (Good, Fair, or Poor) for each study, or at least a summary of these ratings and how any disagreements between raters were resolved.

The NIH National Heart, lung and blood institute Quality Assessment Tool for Observational Cohort and Cross-sectional studies and Case Control studies is an established and widely used quality assessment tool. It was deemed appropriate because all included studies employed an observational study design, to which this quality assessment tool is applicable. Our responses to each of the 14 criteria are detailed in Table 3, followed by the criteria themselves, which allow the reader to verify our ratings to each of the criteria. The criteria on the NIH Quality Assessment Tool are designed to help researchers focus on the key concepts for evaluating the internal validity of a study. They are not intended to create a list that could be tally up to arrive at a summary judgment of quality.

We have revised the Data Extraction and Quality Assessment paragraph in Methods section to clarify this further for the reader.

The results are nicely organized and succinct. They would benefit from a Table summarizing the specific findings from each study in more detail.

We have added such a table in Appendix 3.

One concern is the heterogeneity of the studies (which the authors also point out), with variability in the original research questions, clinical settings, country in which the study was conducted, and health care systems. It would be useful, therefore, to have each result contextualized by this information, particularly when results are from a single study, which could make them appear more generalizable than they are. For example, the result that "seeking a SO was common in non-religious patients vs. religious patients" would be enhanced with information about the setting (pediatric oncology) and country (Israel).

Thank you for this comment. We have revised the following paragraph to enhance with information about the setting in the "Patient Characteristics associated with SO Seeking" paragraph.

"Seeking a SO was more common in non-religious patients vs. religious patients having cancer in Israel [42], in patients having cancer who were employed in the US [37,39] and in patients with higher income and socioeconomic status [12,30,31,36,42]. SOs were more common among breast cancer patients who had private insurance [37], and among men with localised prostate cancer with a private insurance in the US [39]. Two studies reported on geographic residency, more common for those living in central areas in Israel [12] and for those closer to a SO centre in the Netherlands [47].

Patients seeking a SO with breast cancer were more actively involved in decision-making processes in Germany [40]. Patients seeking a SO from Orthopaedics had a poorer relationship with their first doctor in the Netherland [47], and those seeking SO in Japan were more anxious and believed they were in poor health [28].”

I also question the accuracy of some of the results. For example, I struggled to reconcile the findings in the manuscript with the Annadale and Hunt (1998) study. On p. 8, lines 39-41, there is a reported age result in the cancer setting. The original paper, however, drew on a sample of participants from a 35-year-old age cohort, there were no age-related findings, and there were a variety of health complaints. Further, in Table 1 this study is reported as having 136 participants, when in fact there were 307 cases that included disagreements, of which 136 reported seeking a second opinion.

Thank you for this comment. The details of this study were rechecked and corrected in the table and text.

How patients are engaged in decision making is a focus of much current scholarship, and the authors have undertaken an ambitious review related to this important topic: why do patients seek second opinions, under what conditions do they do so, and what outcomes result? The manuscript would benefit from increased rigor in method and accuracy of results, which could then contribute a valuable update to the 2014 systematic review.

We hope that we have now responded to the reviewer’s concerns relating to the specific elements mentioned.

Reviewer: 4

Dr. Yan Chen, University of Auckland

Comments to the Author:

This paper is well written and describes the study methods and findings in great detail. I was invited to review the paper with a particular emphasis on its statistical methods and analyses. The manuscript follows the PRISMA guideline and provides sufficient information to adhere to the guideline. I have very few minor comments regarding the study’s analyses, given that it has demonstrated sound methodology.

In the eligibility criteria, the authors excluded subspecialized domains such as dentistry and psychiatry; whereas some other specialities such as neurology, ophthalmology, and orthopaedics

were included. It would have been beneficial to explain why psychiatry, as a medical sub-specialty, was being excluded.

We would like to thank you for taking the time to review the paper and provide helpful comments, which improve the quality for manuscript.

This review focused on medical and surgical SOs and as such included medical and surgical subspecialties. Dentistry was considered to fall outside this broad definition. Whilst psychiatry is of course a medical subspecialty, it was felt to be sufficiently distinct from other areas of medicine as to be better excluded from this review. Further, it was felt that the particular nature of psychiatry and of patients requiring the services of psychiatric professionals meant that the motivations for SO seeking might reasonably be expected to differ significantly to those in other fields and consequently unnecessarily complicate or bias the generalisable findings of this review.

There are some inconsistencies in-text. For instance, the authors referred to Appendix 2 in-text, while I assume that they were actually referring to Table 2 instead.

The tables and appendixes numbering has been corrected.

VERSION 2 – REVIEW

REVIEWER	Gerwing, Jennifer Akershus University Hospital, Health Services Research
REVIEW RETURNED	01-Jul-2021
GENERAL COMMENTS	I'm satisfied with the authors' responses to my review and revisions to the manuscript